# The Antibacterial Properties of Nanocomposites Based on Carbon Nanotubes and Metal Oxides Functionalized with Azithromycin and Ciprofloxacin

**DOI:** 10.3390/nano12234115

**Published:** 2022-11-22

**Authors:** Adina Stegarescu, Ildiko Lung, Alexandra Ciorîță, Irina Kacso, Ocsana Opriș, Maria-Loredana Soran, Albert Soran

**Affiliations:** 1National Institute for Research and Development of Isotopic and Molecular Technologies, 67-103 Donat, 400293 Cluj-Napoca, Romania; 2Faculty of Biology and Geology, Babeș-Bolyai University, 5-7 Clinicilor, 400006 Cluj-Napoca, Romania; 3Department of Chemistry, Supramolecular Organic and Organometallic Chemistry Centre (SOOMCC), Faculty of Chemistry and Chemical Engineering, Babeş-Bolyai University, 11 Arany Janos, 400028 Cluj-Napoca, Romania

**Keywords:** carbon nanotubes, nanocomposites, magnetite, metal oxides, antibiotics, antibacterial activity, bacteria strains

## Abstract

Different microorganisms are present in nature, some of which are assumed to be hazardous to the human body. It is crucial to control their continuing growth to improve human life. Nanomaterial surface functionalization represents a current topic in continuous evolution that supports the development of new materials with multiple applications in biology, medicine, and the environment. This study focused on the antibacterial activity of different nanocomposites based on functionalized multi-walled carbon nanotubes against four common bacterial strains. Two metal oxides (CuO and NiO) and two antibiotics (azithromycin and ciprofloxacin) were selected for the present study to obtain the following nanocomposites: MWCNT-COOH/Antibiotic, MWCNT-COOH/Fe_3_O_4_/Antibiotic, and MWCNT-COOH/Fe_3_O_4_/MO/Antibiotic. The present study included two Gram-positive bacteria (*Staphylococcus aureus* and *Enterococcus faecalis*) and two Gram-negative bacteria (*Escherichia coli* and *Pseudomonas aeruginosa*). Ciprofloxacin (Cip) functionalized materials (MWCNT-COOH/Fe_3_O_4_/Cip) were most efficient against all tested bacterial strains; therefore, we conclude that Cu and Ni reduce the effects of Cip. The obtained results indicate that the nanocomposites functionalized with Cip are more effective against selected bacteria strains compared to azithromycin (Azi) functionalized nanocomposites. The current work determined the antibacterial activities of different nanocomposites and gave fresh insights into their manufacture for future research regarding environmental depollution.

## 1. Introduction

Carbon nanotubes (CNTs) have extraordinary physicochemical, electrical, and mechanical properties as nanomaterials. Due to their potential in the development of new applications, CNTs are an important material for multiple applications in biology, medicine, and environmental depollution [1]. Single-walled CNTs (SWCNTs) and multi-walled CNTs (MWCNTs) have gained significant attention as supporting materials due to their exceptional properties, including a high surface area and chemical stability [2]. Additionally, MWCNTs have captured interest in the biomedical field, electrochemical applications, and aerospace [3,4]. Furthermore, a very important characteristic of CNTs is their biocompatibility, which contributes to the material’s reduced cellular cytotoxicity [5,6]. Nanoparticles alone are known to possess properties against Gram-positive and Gram-negative bacteria (e.g., *Staphylococcus aureus*, *Bacillus subtilis*, *Listeria monocytogenes*, *Escherichia coli*, *Salmonella typhimurium*, *Pseudomonas aeruginosa*, etc.), such as when used as antibacterial coating materials for some implantable devices, materials for medical use in infection prevention and wound healing, in antibiotic delivery systems to treat disease, in bacterial detection systems to generate microbial diagnostics, and in antibacterial vaccines to control bacterial infections [7].

The decoration of CNTs with metal nanoparticles (NPs) has been demonstrated to have potential applications as a new hybrid material regarding an increase in electrical conductivity and gas sensors [8,9]. Carbon nanotubes (CNTs), due to their ultra-high surface area, have a high mechanical strength but are ultra-light in weight and have excellent chemical and thermal stability, which are considered ideal properties for drug delivery; however, to improve these properties, metal or metal oxide (MO) nanoparticles with magnetic properties can be attached to the surface of CNTs. Referring to previous studies, CNT composites with metal and metal oxide (MO) NPs exhibited excellent antimicrobial activities [10,11,12]. The antibacterial properties of the MO NPs have been advantageous because of their nano-aspect and broad surface area to volume ratio, which increases direct interactions with microbial membranes [13]. The mechanism of MO NPs as antibacterial agents is not only dependent on the chemical composition, but also on the size, shape, agglomeration degree, solubility, and surface charge of the NPs [14]. One such example is the complex with Fe_3_O_4_; by combining the mechanical, electrical, and thermo-optical properties with the superparamagnetic behavior of Fe_3_O_4_, a high strength magnetic material can be produced for use in applications in medicine, electrical devices, magnetic data storage systems, and as heterogeneous catalysts. Magnetic nanoparticles also have the advantage that they can be recovered and reused. Hussain et al. demonstrated that the ZnO composite of CNTs showed more antibacterial potential as compared to ZnO NPs. The antibacterial activity of the combination CNTs/ZnO composite was more efficient against *Bacillus cereus (B. cereus)*, *Escherichia coli* (*E. coli*), and *Pseudomonas aeruginosa* (*P. aeruginosa*), as compared to the ZnO NPs [15]. Another study combined CNTs and other nanocomposites (e.g., cellulose (cotton) nanocomposite) with silver NPs to obtain nanomaterials that demonstrated increased antibacterial activity against two strains of infectious Gram-negative bacteria (*E. coli*) and Gram-positive (*Staphylococcus aureus*—*S. aureus*) [16,17]. More recently, the antimicrobial properties of CuO NPs have been studied, which demonstrated exceptional bactericidal and fungicidal activity [18,19].

The presence of antibiotics in the aquatic environment is the result of both irresponsible use and high demand due to COVID-19 bacterial-associated infections. This is problematic for various reasons, including the acquisition of antibiotic resistance mechanisms in bacteria or the decimation of environmental bacterial strains responsible for organic matter degradation [20]. Azithromycin is a macrolide family antibiotic that acts against a wide variety of bacterial species; it is also known for its antiviral and immunomodulatory activities. The effects of azithromycin against COVID-19 were shown in recent studies [21]. Ciprofloxacin is known for its effectiveness against Gram-negative bacteria; however, it can also inhibit the development of Gram-positive strains [22]. This antibiotic has FDA approval to treat a plethora of infections, which could lead to high accumulation in the water environment [23].

CNTs functionalized with various compounds (i.e., antioxidants, flavonoids, antibiotics) are suitable for water decontamination, as well as disinfection. This type of nanocomposite was proven to be efficient in bacterial inactivation [24]. Kadurugamuwa et al. demonstrated that NPs labeled with gentamicin were active against *P. aeruginosa,* suggesting a substitute mechanism of action that can be available to the aminoglycoside antibiotics class [25]. Other studies demonstrated that a composite based on SWNTs covalently bound to a polyamide membrane was active against *E. coli*, and a hybrid material based on MWCNTs with a coating of immobilized amoxicillin presented antibacterial activity against *E. coli* and *S. aureus* [26,27]. Rotella et al. demonstrated that antibiotic-labeled graphitic carbon nanofibers (GCNFs) covalently functionalized with aminoglycosides (tobramycin and amikacin) and fluoroquinolone (ciprofloxacin) possess significant antibacterial activity against pathogenic organisms, such as *P. aeruginosa* [28,29].

The aim of the present research was to study the antibacterial activity of some nanocomposites (MWCNT-COOH/Antibiotic, MWCNT-COOH/Fe_3_O_4_/Antibiotic, and MWCNT-COOH/Fe_3_O_4_/MO/Antibiotic) against four common pathogenic organisms, including the following: *S. aureus* (ATCC: 25923), *Enterococcus faecalis* (*E. faecalis*, ATCC: 29212), *E. coli* (ATCC: 25922), and *P. aeruginosa* (ATCC: 27853). The first step was to add the magnetite to obtain magnetic nanocomposites followed by the metal oxides (CuO and NiO) to obtain the specified nanocomposites. To obtain a new system, with strong antibacterial activity that could be the solution for water purification and useful for medical purposes, we retained an antibiotic on the surface of these nanocomposites based on carbon nanostructures and investigated the antibacterial properties. For this purpose, ciprofloxacin (Cip) and azithromycin (Azi) were chosen due to them being well-known classes of antibiotics and their applicability against both Gram-positive and Gram-negative bacteria. This antibacterial activity is an interesting prospect for further investigation of nanocomposites based on carbon nanostructures and for potential use in biomedical and environmental applications.

## 2. Materials and Methods

For the synthesis of nanocomposites, MWCNT (D × L 110–170 nm × 5–9 µm), NiCl_2_ × 6H_2_O, CuSO_4_, ascorbic acid, and cetyltrimethylammonium bromide (CTAB) were purchased from Sigma-Aldrich (Schnelldorf, Germany), FeCl_3_ × 6H_2_O from Alfa Aesar (Kandel, Germany), FeSO_4_ × 7H_2_O from VWR Chemicals (Wien, Austria) and NH_3_ 25% solution from Chemical Company (Iași, Romania). Ciprofloxacin and azithromycin, chosen as antibiotics in this study, were purchased from Sigma-Aldrich (Schnelldorf, Germany) and TCI (Japan), respectively. pH adjustment was performed with HCl and NaOH, which were purchased from Sigma-Aldrich (Schnelldorf, Germany) and VWR Chemicals (Wien, Austria), respectively. Aqueous solutions were prepared using ultrapure water (Direct-Q^®^ 3 UV Water Purification System, Merck, Darmstadt, Germany). To determine the antibacterial effect, the diffusion method with antibiogram discs was employed. The nanocomposites and antibiotics (25 mg/mL, 10 µL/disc) were placed on Ø 6 mm Wattmann discs and left to interact with the bacterial strains for 24 h. The tested bacterial strains were as follows: *Staphylococcus aureus* (ATCC: 25923), *Enterococcus faecalis* (ATCC: 29212), *Escherichia coli* (ATCC: 25922), and *Pseudomonas aeruginosa* (ATCC: 27853). The experiments were performed in duplicate and in accordance with EUCAST protocols [30,31,32]. The characterization of nanocomposites was performed by Fourier-transform infrared spectroscopy (FTIR) using a JASCO 6100 FTIR spectrometer (Tokyo, Japan). FTIR spectra were recorded in the spectral range of 4000 to 400 cm^−1^, with a resolution of 4 cm^−1^, using the KBr pellet technique. The collected spectra were analyzed with Jasco Spectra Manager v.2 software, a soft Spectra Manager Version 2.05.03, copyright 2002–2006, Jasco Corporation, Tokyo, Japan while the morphological characterization of the samples was performed by electron microscopy measurements in transmission using a STEM HITACHI HD-2700 microscope.

## 3. Synthesis of Nanocomposites

### 3.1. MWCNT-COOH/Fe_3_O_4_ Synthesis

The MWCNT functionalized with COOH groups and MWCNT-COOH/Fe_3_O_4_ nanocomposites were synthesized according to the methods published in our previous papers [33,34,35].

### 3.2. MWCNT-COOH/Fe_3_O_4_/NiO or CuO Synthesis

The synthesis of MWCNT-COOH/Fe_3_O_4_/NiO has been reported previously [34]. To obtain the MWCNT-COOH/Fe_3_O_4_/CuO nanocomposite, a similar synthesis chain was performed as follows: the mixture of CNT-COOH/Fe_3_O_4_ was sonicated for 30 min in water at a ratio of 1.66:1 (*w*/*v*). Next, freshly prepared solutions of 0.3747 g CuSO_4_ × 5H_2_O in 50 mL of water and 0.9687 g of ascorbic acid in 50 mL of water were added. After the addition of 0.5467 g of CTAB to the previously obtained suspension, another 50 mL of water was added and the whole mixture was pH adjusted to 6.5 using NaOH solution. The solution was then heated to 85 °C and further stirred for another 3 h. Finally, the sample was washed with water by centrifugation and dried in an oven at 75 °C.

### 3.3. Synthesis of Nanocomposites MWCNT-COOH/Antibiotic, MWCNT-COOH/Fe_3_O_4_/Antibiotic, and MWCNT-COOH/Fe_3_O_4_/MO/Antibiotic

The nanocomposite (0.1 g) was dissolved in 30 mL tetrahydrofuran, followed by the addition of 0.025 g of antibiotic. The obtained mixture was stirred for 2 days on a magnetic stirrer at room temperature. The mixture was then washed by centrifugation four times with tetrahydrofuran using a 30 mL/wash cycle. The nanocomposites obtained in this way were dried in an oven for 12 h at 75 °C. The schematic reactions for obtaining nanocomposites can be observed in Figure 1.

## 4. Results and Discussion

### 4.1. Morphological Characterization

Transmission electron microscopy (TEM) images of the obtained nanocomposites are shown in Figure 1. The images show that the nanoparticles are not agglomerated or adhered to each other, which suggests that the materials have a large surface area for reaction. In the case of MWCNT-COOH/Fe_3_O_4_/NiO/Cip and MWCNT-COOH/Fe_3_O_4_/CuO/Cip nanocomposites, it can be observed that the Fe_3_O_4_ spherical nanoparticles are covered on the surface with MO (NiO or CuO), which indicates that the prepared nanocomposites are stable in the composite. Some of the nanoparticles were agglomerated because of the magnetic properties of the developed nanocomposite. Such agglomeration can be explained by the high iron content of magnetite nanoparticles, which led to their agglomeration. The same behavior was observed in the case of nanocomposites functionalized with azithromycin. The presence of ciprofloxacin or azithromycin on the nanocomposite surface could be determined only by IR spectroscopy.

### 4.2. FTIR Analysis

To assess the attachment of the antibiotic on the nanocomposite surface, IR spectroscopy was performed. The spectra of the functionalized nanotubes with –COOH groups, MWCNT-COOH, show the following characteristic vibration bands in the 1800–1000 cm^−1^ spectral domain: at 1690 cm^−1^ the stretching vibration of the –COOH group, at 1582 cm^−1^ the stretching vibration of the C=O bonds linked by hydrogen bridges, and at 1528 and 1162 cm^−1^ the stretching vibrations of the C=C and C-C-C bonds in the structure of the carbon nanotubes [36,37]. These vibration bands appear slightly shifted in the MWCNT–COOH/Fe_3_O_4_ spectrum as follows: 1725, 1613, 1567, and 1201 cm^−1^, respectively, which is probably due to the interaction between the magnetic nanoparticles and the surface of the functionalized nanotubes. The characteristic absorption band of Fe-O bonds from Fe_3_O_4_ appears in the 375–650 cm^−1^ range, with the maximum at 596 cm^−1^ [38]. In the 1800–1000 cm^−1^ spectral domain of the MWCNT–COOH/Fe_3_O_4_/CuO spectrum, the vibration bands attributed to C=O groups linked by hydrogen bonds and C=C and C-C-C bonds appear slightly shifted to 1746, 1625, 1575, and 1213 cm^−1^, respectively. The characteristic band for the Fe-O bond, with a lower intensity, appears at 594 cm^−1^. The presence of CuO on the surface shows vibration bands from 615, 510, and 450 cm^−1^ [39,40].

In the case of the MWCNT–COOH/Fe_3_O_4_/NiO, in the 1800–1000 cm^−1^ range, the following vibration bands can be identified: 1694, 1604, 1486, 1465, 1384, 1361, and 1316 cm^−1^. The characteristic band of the Fe-O bonds, which is of low intensity, appears at 595 cm^−1^. The appearance of a new band at 428 cm^−1^ indicates the presence of Ni–O on the structure of the nanoparticles [41].

Following functionalization of the nanocomposites with ciprofloxacin, the characteristic absorption bands can be identified on the spectra in the case of all four types of nanotubes; however, some of them are slightly shifted as follows: 3285 cm^−1^ (-OH), 1707 cm^−1^ (–COOH), 1625 cm^−1^ (–C=O), 1493 cm^−1^, 1467 cm^−1^ (-C-H), 1449 cm^−1^ (–C-O ), 1383 cm^−1^ (-C=C aromatic), and 1271 cm^−1^ (-C-F) [42,43], which provides evidence of its presence on the surface of the nanotubes exposed to the treatment (Figure 2, Figure 3 and Figure 4). The displacement of some vibration bands and a change in intensity and broadening of the absorption bands from 1707, 1625, 1383, and 1271 cm^−1^ can be observed. The displacement of the bands and their broadening was most pronounced for the MWCNT–COOH/Fe_3_O_4_/Cip and MWCNT–COOH/Fe_3_O_4_/NiO/Cip samples.

The most important characteristics of the vibration bands for azithromycin, in accordance with the literature [44,45,46], are as follows: 3558 and 3485 cm^−1^ (O-H); 2971 and 2783 cm^−1^ (C-H); 1720 cm^−1^ (-C=O); 1638 cm^−1^, 1458 cm^−1^, 1378 cm^−1^, 1343 cm^−1^, 1281 cm^−1^, and 1185 cm^−1^ (-C-O); 1082 cm^−1^ and 1051 cm^−1^ (R-O-R and C-N), 995 cm^−1^, 900 cm^−1^, 834 cm^−1^, 795 cm^−1^, and 730 cm^−1^ (C-H), 639 cm^−1^ and 573 cm^−1^.

The presence of azithromycin on the surface of the nanocomposite can be observed on the spectra of all four types of nanotubes (Figure 5, Figure 6 and Figure 7). Some of the most important vibrational bands are slightly shifted and broadened as follows: the C=O stretching vibration is shifted from 1720 to 1744 cm^−1^, the absorption bands from 1458, 1378, and 1051 cm^−1^ are shifted to 1744, 1463, 1390, and 1041 cm^−1^, respectively. The displacements of these bands and their intensities are the most pronounced for the MWCNT–COOH/Fe_3_O_4_/NiO-Azi sample.

### 4.3. Antibacterial Activity

The antibacterial effects of the obtained nanocomposites were assessed through the antibiogram method (Figure 8, Table 1). *S. aureus* was susceptible to MWCNT-COOH/Fe_3_O_4_/CuO and MWCNT-COOH/Fe_3_O_4_/NiO, indicating that the effect observed on the composites that had ciprofloxacin is also due to the Fe, Cu, and Ni elements. *E. coli* was also susceptible to MWCNT-COOH/Fe_3_O_4_/CuO and MWCNT-COOH/Fe_3_O_4_/NiO, but only in one out of the two replicates. The average inhibition was 7.5 and 8 mm, respectively; however, the standard deviation was over 10 mm due to the resistance observed in one of the replicates. All other composites without ciprofloxacin had no antibacterial effect against the tested strains.

Regarding the ciprofloxacin-functionalized materials, MWCNT-COOH/Fe_3_O_4_/Cip was most efficient against all tested bacterial strains (Table 2). The remaining composites had the following degrees of efficiency for *E. coli* and *S. aureus*: MWCNT-COOH/Cip > MWCNT-COOH/Fe_3_O_4_/CuO/Cip > MWCNT-COOH/Fe_3_O_4_/NiO/Cip, and for *P. aeruginosa* and *E. faecalis*: MWCNT-COOH/Cip > MWCNT-COOH/Fe_3_O_4_/NiO/Cip > MWCNT-COOH/Fe_3_O_4_/CuO/Cip. These results indicate that Cu and Ni downsize the effects of ciprofloxacin.

For the azithromycin functionalized samples, the most effective nanocomposites were MWCNT-COOH/Azi and MWCNT-COOH/Fe_3_O_4_/CuO/Azi, which inhibited all bacterial samples to different degrees (Table 2). MWCNT-COOH/Fe_3_O_4_/Azi had mild effects against *S. aureus* (~8 mm Ø), while MWCNT-COOH/Fe_3_O_4_/NiO/Azi affected both *S. aureus* and *E. faecalis* (~8.5 mm Ø).

The obtained results indicate that the nanocomposites functionalized with Cip are more effective against Gram-positive and Gram-negative bacteria, compared to the Azi-functionalized nanocomposites. In addition, as compared to antibiotics alone, in all cases, the Cip-functionalized nanocomposites had greater inhibitory capacities. However, in the case of Azi-functionalized nanocomposites, the response was generally weaker as compared to the antibiotic alone (Table 1 and Table 2). There are four possible mechanisms for bacterial inhibition that can explain the findings in the current study. First, regarding the carbon nanotubes, their mode of action at subcellular levels is still under investigation; however, their antibacterial properties might be more related to the surface charge and hydrophobicity properties that in preventing the attachment of bacteria [1]. Second, the Fe_3_O_4_ NPs are known to produce reactive oxygen species (ROS), which have negative effects on bacteria, inhibiting their development [47]. Third, regarding Cu/Ni NPs doping of the MWCNT, Cu and Ni are materials well known for their cytotoxicity and capacity to inhibit bacterial growth [48,49]. There are multiple modes of action for Cu and Ni, including disruption of the outer membrane and the induction of ROS [48]. Fourthly, the antibiotics can be adsorbed on the surface of the MWCNTs.

## 5. Conclusions

New nanocomposites based on carbon nanostructures and metal nanoparticles and/or oxide nanoparticles functionalized with antibiotics (ciprofloxacin and azithromycin) were obtained. Morphological characterization of these new nanocomposites was performed using TEM analysis, and the presence of the antibiotics on the surface of the nanocomposites was confirmed by FTIR analysis. These new nanocomposites were prepared and tested for antibacterial activity. The obtained powdered nanocomposites were incubated with the four most common bacterial strains: *Pseudomonas aeruginosa* (ATCC: 27853), *Escherichia coli* (ATCC: 25922), *Staphylococcus aureus* (ATCC: 25923), and *Enterococcus faecalis* (ATCC: 29212). Among the materials functionalized with antibiotics, the most effective was MWCNT-COOH/Fe_3_O_4_/Cip against all strains used, as compared to the azithromycin functionalized nanocomposites. Further studies are needed to evaluate the applicability of synthesized nanocomposites in medicine and environmental depollution.

## Data Availability

The data presented in this study are available on request from the corresponding author.

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
