# Peer review of "The Antibacterial Properties of Nanocomposites Based on Carbon Nanotubes and Metal Oxides Functionalized with Azithromycin and Ciprofloxacin"

_nanomaterials, 2022, doi:10.3390/nano12234115_

Round 1
Reviewer 1 Report
Journal: Nanomaterials
Ms. ID.: nanomaterials-2039907
Title: The antibacterial properties of nanocomposites based on carbon nanotubes and metal oxides functionalized with azithromycin and ciprofloxacin
Stegarescu et al. reported a study on the antibacterial activity of some nanocomposites against four common pathogenic organisms S. aureus, Enterococcus faecalis, E. coli, and P. aeruginosa. The oxides used for obtaining the specified nanocomposites were CuO and NiO, and the antibiotics used were ciprofloxacin (Cip) and azithromycin (Azi). The magnetite was added to obtain magnetic nanocomposites. The authors claim that the results obtained within this study could indicate that nanocomposites with antibacterial activity could be solutions for water purification and their use for medical purposes. It is a very interesting manuscript. It fits well with the scope of the Journal. I consider the manuscript suitable for publication, but I also believe some important improvements are needed. The list of specific issues that should be addressed is listed below.
-Line 35: “Due to their potential in developing new applications, CNTs are essential” – it is a strong statement. Please modify.
-the Introduction is generally ok, but the motivation for synthesizing antibiotics composites is not obvious. It should be more pronounced. Also, the novelty of the manuscript should be clear.
-FTIR is the only technique used to confirm the presence of the antibiotics on the surface of the nanocomposites in this manuscript. It would be highly beneficial to use some other technique as well, for example, XPS.
-antibacterial activity testing seems inconclusive since there is no information about the influence of Cip and Azi on the tested strains alone. It should be included in the study, and the discussion should be rewritten with respect to the new information obtained.
Author Response
Thank you for the review and for the suggestions!

Reviewer 2 Report
Metal and metal oxide nanoparticles show promising application in antibacterial fields. The topic of this manuscript is of broad interest to the readers. However, major revision is required before acceptation.
1. What is “MO” in line 19? Please give the full name.
2. “Figure 1. Representative TEM images of the obtained nanocomposites.” should be revised. The exact name of the nanocomposites should be given since a few composites have been prepared.
3. Please pay attention to the figure number. There are two Figure 1.
4. The wavenumber “1/cm” in X axial of IR spectra need to be revised as “cm-1”.
5. XRD patterns of the nanocomposites are suggested to be added to reveal the crystal structure.
6. The authors should pay attention to the writing of units. There should be a space between the unit and the number.
7. Most of the references are too old which cannot represent the update progress in the field. More references published in recently years are suggested to be added, for example Journal of Bioresources and Bioproducts 2021, 6 (1), 75-81; Journal of Bioresources and Bioproducts 2021, 6 (1), 26-32; Journal of Bioresources and Bioproducts 2020, 5 (3), 180-185.
Author Response

(The authors gave the same response as above.)

Round 2
Reviewer 1 Report
The authors addressed all my comments.
Reviewer 2 Report
The manuscript has been revised according to the comments and could be accepted.